# An Unbiased Immunization Strategy Results in the Identification of Enolase as a Potential Marker for Nanobody-Based Detection of *Trypanosoma evansi*

**DOI:** 10.3390/vaccines8030415

**Published:** 2020-07-24

**Authors:** Zeng Li, Joar Esteban Pinto Torres, Julie Goossens, Didier Vertommen, Guy Caljon, Yann G.-J. Sterckx, Stefan Magez

**Affiliations:** 1Laboratory for Cellular and Molecular Immunology (CMIM), Department of Bioengineering Sciences, Vrije Universiteit Brussel, B-1050 Brussels, Belgium; zeng.li@vub.be (Z.L.); joar.pinto@vub.be (J.E.P.T.); goossensjulie@hotmail.com (J.G.); 2Laboratory of Medical Biochemistry (LMB) and the Infla-Med Centre of Excellence, Department of Pharmaceutical Sciences, University of Antwerp, B-2610 Wilrijk, Belgium; Yann.sterckx@uantwerpen.be; 3MASSPROT Platform, de Duve Institute, Université Catholique de Louvain, B-1200 Brussels, Belgium; didier.vertommen@uclouvain.be; 4Laboratory of Microbiology, Parasitology and Hygiene (LMPH) and the Infla-Med Centre of Excellence, Department of Biomedical Sciences, University of Antwerp, B-2610 Wilrijk, Belgium; Guy.Caljon@uantwerpen.be; 5Department of Biochemistry and Microbiology, Ghent University, B-9000 Gent, Belgium; 6Laboratory for Biomedical Research, Department of Molecular Biotechnology, Environment Technology and Food Technology, Ghent University Global Campus, Incheon 406-840, Korea

**Keywords:** *Trypanosoma evansi*, Nanobody, enolase, diagnosis, homologous and heterologous detection assays

## Abstract

*Trypanosoma evansi* is a widely spread parasite that causes the debilitating disease “surra” in several types of ungulates. This severely challenges livestock rearing and heavily weighs on the socio-economic development in the affected areas, which include countries on five continents. Active case finding requires a sensitive and specific diagnostic test. In this paper, we describe the application of an unbiased immunization strategy to identify potential biomarkers for Nanobody (Nb)-based detection of *T. evansi* infections. Alpaca immunization with soluble lysates from different *T. evansi* strains followed by panning against *T. evansi* secretome resulted in the selection of a single Nb (Nb11). By combining Nb11-mediated immuno-capturing with mass spectrometry, the *T. evansi* target antigen was identified as the glycolytic enzyme enolase. Four additional anti-enolase binders were subsequently generated by immunizing another alpaca with the recombinant target enzyme. Together with Nb11, these binders were evaluated for their potential use in a heterologous sandwich detection format. Three Nb pairs were identified as candidates for the further development of an antigen-based assay for Nb-mediated diagnosis of *T. evansi* infection.

## 1. Introduction

*Trypanosoma evansi* is the most widely spread animal-infective trypanosome and causes a disease known as “surra”, which affects both domestic livestock and wildlife (e.g., horses, cattle, camels, buffaloes, pigs, and deer) [1,2,3]. The pathology is characterized by weight loss, drastic reductions of draft power, diminished meat and milk production, and often leads to death of the infected animals. Hence, this has a significant socio-economic impact on the local cattle industry in the affected areas [4,5]. While the transmission of most salivarian trypanosomes depends on the tsetse fly, *T. evansi* has extended its vector range to a number of hematophagous biting flies other than tsetse. This has allowed *T. evansi* to spread beyond the “tsetse belt” and move out of Africa. Today, *T. evansi* is found in South America, Africa, Asia, and even Oceania [6,7]. Occasionally, surra outbreaks threaten Europe, when undiagnosed infected animals (camels, equids, and dogs) are brought into the continent and allow the parasite to spread to new hosts [8]. Although rare, recent reports have described human *T. evansi* infections, which could indicate that *T. evansi* may be an emerging human pathogen [9,10]. Compared to tsetse-transmitted trypanosomosis, the containment of “surra” is challenging because of the lack of vector specificity displayed by *T. evansi*. Due to these issues, disease control is largely based on the use of trypanocides and preventive management methods [11]. In principle, drug treatment in livestock should ideally be combined with a conclusive diagnosis of infection or cure. Therefore, establishing an effective detection method for active infections is important in formulating appropriate strategies to control *T. evansi*.

*T. evansi* strains are categorized into one of two groups depending on the presence (Type A) or absence (Type B) of the gene encoding the RoTat1.2 variant surface glycoprotein (VSG) and/or variation of their kinetoplast minicircle DNA [12,13,14]. The existing *T. evansi* diagnostic methods include parasitological, nucleic acid-based, and immunological assays that allow detection of Type A and/or B strains depending on the test. The parasitological techniques are based on direct parasite observation using light microscopy but have the drawback of possessing a low sensitivity and only being effective in the acute phase of the disease when the parasitemia tends to be relatively high. Nucleic acid-based assays such as polymerase chain reaction (PCR), loop-mediated isothermal amplification (LAMP), or recombinase polymerase amplification (RPA) consist of amplifying sequences specific to the *T. evansi* genome and are effective to confirm active infections [15,16,17,18]. Unlike nucleic acid-based techniques, immunological methods detect the presence of specific molecules at the protein level and several immunoassays have been developed for *T. evansi* detection [19,20,21]. Immunoassays essentially fall under one of two types; i.e., antibody-based and antigen-based assays. While the former employ native or recombinant parasite antigens to capture circulating host antibodies, the latter make use of antibodies or fragments thereof for the detection of circulating parasite antigens. Antigen-based assays typically make use of an antibody pair consisting of so-called capturing and detecting antibodies. These can either recognize the same epitope on the target antigen or can bind distinct epitopes, in which case the assay is called homologous or heterologous, respectively. While homologous antigen-based assays require the target antigen to be a multimer (or a monomer with repeating epitopes), this is not a necessity for heterologous assays [22,23,24]. Importantly, in both formats the capturing and detection antibodies should be able to outcompete infection-induced anti-parasite host antibodies [25], and must neither cross-react with host anti-IgG autoantibodies [26]. These issues can be circumvented by the use of camelid single-domain antibodies, otherwise known as Nanobodies (Nbs) [27,28,29,30].

Nbs are single-domain antibody fragments that correspond to the variable antigen-binding domain (VHH) of camelid heavy-chain only antibodies (HCAbs) [31]. They represent the structural and functional equivalent of the F_ab_ fragment of conventional antibodies [32]. Despite the loss of the variable light chain in HCAbs, VHHs retain the capacity to recognize and bind epitopes with large surface areas due to the occurrence of longer antigen-binding loops (complementarity determining regions, CDRs) compared to the those of conventional antibodies (especially true for CDRs 1 and 3) [33]. As a result, Nbs possess a convex paratope with extending CDRs that allow them to reach epitopes that often cannot be easily accessed by conventional antibodies (such as clefts and cavities on the antigen surface) [34,35]. The combination of their ability to recognize cryptic epitopes, their small size (≈15 kDa for a Nb vs. ≈150 kDa for a conventional antibody), high thermostability and solubility, ease of production in bacteria and the high-throughput identification of potent binders via display technologies and biopanning procedures offers many advantages in the development of diagnostic and therapeutic applications [31,36,37,38].

In this paper, we describe the application of an unbiased immunization strategy to identify potential biomarkers for Nb-based detection of a wide range of *T. evansi* strains (both Type A and B). The approach consists of immunizing a camelid with a heterogeneous antigen mixture of unknown compositions (e.g., parasite secretome or soluble lysate) to construct a Nb library which is then panned against the same antigen mix. The identified Nbs are then used to determine the nature of their target antigens via immunocapturing followed by mass spectrometry (MS). The approach is “unbiased” given that Nb library construction is performed without prior knowledge of the target antigen(s) recognized by the identified binders [24,39]. Here, we report the identification of a single anti-*T. evansi* secretome Nb (Nb11) after immunizing an alpaca with soluble lysate preparations from different *T. evansi* strains followed by panning and screening against the *T. evansi* secretome. Using subsequent Nb11-mediated immunocapturing and MS, the Nb11 target was identified as the glycolytic enzyme *T. evansi* enolase (*Tev*ENO). A combination of analytical gel filtration and ELISA shows that Nb11 can be employed for the development of an antigen-based homologous immunoassay targeting *Tev*ENO. Finally, immunization of a second alpaca with recombinantly produced *Tev*ENO yielded four additional anti-*Tev*ENO Nbs, of which three can be used in conjunction with Nb11 to develop heterologous *Tev*ENO sandwich detection assays.

## 2. Materials and Methods

### 2.1. Ethical Statement

All animal experiments were carried out according to the directive 2010/63/EU of the European parliament for the protection of animals used for scientific purposes and approved by the Ethical Committee for Animal Experiments of the Vrije Universiteit Brussel (clearance number 14-220-19 for the alpaca immunization).

### 2.2. Secretome and Soluble Lysate Preparation

#### 2.2.1. Secretome Preparation

Secretome from *T. evansi* strain STIB806 was kindly provided by Dr. Philippe Holzmuller (CIRAD Montpellier, France). Secretome was prepared as previously described [40]. Briefly, isolated parasites from infected rat blood were cultured at a density of 2 × 10^8^ parasites mL^−1^ for 2 h at 37 °C in secretion medium (Ringer Lactate, glucose 0.6%, KCl 0.4%, NaHCO_3_ 0.125%, polymyxin B 5 mg mL^−1^, L-glutamine 2 mM, MEM non-essential amino acids, pH 8.0). Parasite viability was determined by flow cytometry after propidium iodide (PI) staining. Supernatant containing secretome was purified from trypanosomes by centrifugation (1000× *g* for 10 min at 4 °C) and filtered using a 0.22 μm low-binding protein filters. Secretome was aliquoted and stored at −80 °C after addition of the protease inhibitors (30 μg mL^−1^ AEBSF; 1 μg mL^−1^ Leupeptin) until further use.

#### 2.2.2. Soluble Lysate Preparation

Soluble lysates from different *T. evansi* strains (CAN86K, KETRI2479, RoTat 1.2, Merzouga I, STIB816, and AnTaR3) were prepared from parasites isolated from infected mice. Briefly, the BALB/c mice were infected with 10,000 parasites of each *T. evansi* stabilate intraperitoneally. The mice were sacrificed at the parasitemia peak and heparinized blood was collected. Parasites were purified from infected mice blood via DEAE-cellulose as described [41]. After washing three times in phosphate saline glucose pH 8.0 (with Complete^TM^ protease inhibitor, Roche) through centrifugation, parasites were counted under a light microscope and aliquoted at a density of 2 × 10^8^ parasites mL^−1^. Total lysates were prepared by performing three rounds of freezing and thawing followed by five sonication rounds of 10 s each (Soniprep 150, SANYO, Osaka, Japan). After centrifugation at 20,817× *g* for 30 min, the supernatant containing the soluble lysate was collected, aliquoted, and stored at −80 °C until further use.

### 2.3. Nb Library Construction and Phage Display

#### 2.3.1. Construction of anti-*T. Evansi* Lysate and anti-*Tev*ENO Nb Libraries

The anti-*T. evansi* lysate Nb library was constructed by immunizing an alpaca with one of six soluble lysate samples prepared from six different *T. evansi* strains in a sequential manner in order to target conserved epitopes. Every week, 300 µg *T. evansi* soluble lysate prepared in a volume of 500 µL was mixed with an equal volume of GERBU adjuvant (GERBU Biotechnik GmbH, Germany) prior to injection. This process was repeated until the animal had been immunized with the soluble lysates from each of the six selected strains over a course of 6 weeks (week 1: CAN86K, week 2: KETRI2479, week 3: RoTat 1.2, week 4: Merzouga I, week 5: STIB816, week 6: AnTAR3). A total of 50 mL of blood was collected from the alpaca 4 days after the last immunization. Two Nb libraries were constructed as described previously [42].

The anti-*Tev*ENO Nb library was generated through the immunization of a second alpaca with recombinantly produced *Tev*ENO on a weekly basis over a period of 6 weeks. Prior to injection, 100 µg *Tev*ENO prepared in a volume of 500 µL was mixed with an equal volume of GERBU adjuvant. The collection of lymphocytes from the immunized animal’s blood and subsequent library construction were performed as described above.

#### 2.3.2. Phage Display and Panning of the anti-*T. Evansi* Lysate Nb Library

The anti-*T. evansi* lysate Nb library (1 mL) was grown in 300 mL of 2×TY medium supplemented with 100 μg mL^−1^ ampicillin and 2% glucose at 37 °C while shaking at 225 rpm, until OD_600nm_ reached 0.6–0.8. After that, they were transfected with 10^12^ M13K07 helper phages (Invitrogen^TM^) for 30 min at room temperature (RT). Transformants were harvested by centrifugation (1500× *g* for 10 min at 4 °C) and inoculated in 300 mL of fresh 2×TY medium supplemented with 100 μg mL^−1^ ampicillin, 70 μg mL^−1^ kanamycin, and 2% glucose. The culture was grown overnight at 37 °C with shaking at 225 rpm. Finally, cultures were centrifuged (11,000× *g* for 30 min at 4 °C) and the supernatant containing phage particles was collected and mixed with polyethylene glycol (PEG)/NaCl to precipitate the phage particles and resuspended in a final volume of 1 mL. The concentration of phage particles was determined by measuring the OD_260nm_ (OD_260nm_ of 1 = 3 × 10^11^ phages mL^−1^). A total of 1 × 10^11^ phages particles were used to pan against the *T. evansi* secretome (500 μg mL^−1^) coated overnight at 4 °C in the wells of a 96-well NUNC plate (Thermo scientific). The coated wells were blocked with 5% skimmed-milk, protein-free blocking buffer (Thermo scientific), and 0.1% casein in the sequential panning rounds, to minimize the enrichment for binding to “blocking” reagents. Phage particles were eluted with 100 µL of 100 mM triethylamine (pH 11.0) followed by neutralization with 100 μL 1 M Tris-HCl (pH 8.2). Eluted phage particles were amplified by infecting fresh *Escherichia coli* TG1 cells in the exponential phase. Panning of the Nb library was performed for three consecutive rounds on *T. evansi* lysate Nb library.

Screening and selection of colonies carrying Nb fragments targeting the *T. evansi* secretome were performed through an ELISA on periplasmic extracts (PE-ELISA). From the three rounds of panning of the Nb library on *T. evansi* lysate, 190 individual colonies were randomly selected and grown in 100 μL of 2×TY medium supplemented with 10% glycerol, 2% glucose, and 100 μg mL^−1^ ampicillin in 96-well round bottom culture plates. After overnight growing at 37 °C, cells were inoculated in 1 mL of 2×TY medium supplemented with 2% glucose and 100 μg mL^−1^ ampicillin in 96 deep-wells plates. After growing at 37 °C for 4–5 h with shaking at 500 rpm, the culture was induced with 1 mM isopropyl β-D-1-thiogalactopyanoside (IPTG). After growing for another 4 h at 37 °C with shaking at 500 rpm, cells were harvested by centrifugation for 20 min at 3220× *g* at 4 °C. The periplasmic extract containing Nbs was released through freeze-thaw cycles. In parallel, Nunc^TM^ microtiter plates were coated with 500 μg mL^−1^
*T. evansi* secretome and uncoated wells were used as a negative control for each colony. The next day, 100 μL of each periplasmic extract (PE) was loaded in duplicate onto blocked wells and incubated for 1 h at room temperature. The PE-ELISA was detected by mouse anti-hemagglutinin-biotin antibody (anti-HA-biotin, Sigma) diluted in 1:4000 and streptavidin-HRP conjugate antibody (Jackson ImmunoResearch laboratories) diluted in 1:4000. The ELISA was developed by 3,3′,5,5′-tetramethylbenzine (TMB) substrate and stopped by 1:3 addition of 1 M H_2_SO_4_. Based on the OD_450nm_, a true binder was considered when the signal ratio between coated/uncoated wells was ≥2.

#### 2.3.3. Phage Display and Panning of the anti-*Tev*ENO Nb Library

The anti-*Tev*ENO Nb library was panned against *Tev*ENO and *T. evansi* secretome in parallel. Panning was performed as three consecutive rounds on *T. evansi* secretome at concentration of 500 μg mL^−1^ and four consecutive rounds on *Tev*ENO at different concentrations (10 μg mL^−1^ of 1st Round, 5 μg mL^−1^ of 2nd Round, 2.5 μg mL^−1^ of 3rd Round, and 1 μg mL^−1^ of 4th Round, respectively). Screening and selection of colonies carrying *Tev*ENO-targeting Nb fragments was only performed on recombinant protein at concentration of 1 μg mL^−1^ via PE-ELISA. The approach was the same as described above.

### 2.4. Production and Purification of Nbs

The pMECS plasmid containing the Nb gene was extracted from TG1 cells and transformed into *E. coli* WK6 cells for Nb expression. Single colonies were pre-cultured in 5 mL Luria–Bertani (LB) medium supplemented with 100 μg mL^−1^ ampicillin overnight at 37 °C. The next day, 1 mL of each pre-culture was inoculated into a flask containing 330 mL of Terrific Broth (TB) medium supplemented with 2% glucose, 2 mM MgCl_2_, and 100 μg mL^−1^ ampicillin. Cells were grown at 37 °C with shaking at 220 rpm until the OD_600 nm_ reached 0.6–0.8 and gene expression induced by addition of 1 mM IPTG and growing overnight at 28 °C with shaking at 220 rpm. Cells were harvested by centrifugation for 8 min at 11,000× *g* at 4 °C. Pellets were lysed by osmotic shock and the PE was collected by centrifugation for 30 min at 11,000× *g* at 4 °C.

The PE extracts containing Nbs were incubated with His-Select Nickel resin (Sigma) for 1 h at RT with shaking, then passed over a PD-10 column. Proteins were eluted with PBS (pH 7.5) containing 1 M of imidazole and further purified by size exclusion chromatography (SEC) on a Superdex 75 16/60 column (GE Healthcare) which was pre-equilibrated with PBS for at least one column volume. The Nbs were finally eluted with PBS. Fractions containing the Nbs were pooled and stored at 4 °C for further use. Each purification step was monitored by SDS-PAGE under reducing conditions and visualized by Coomassie blue staining.

### 2.5. Immuno-Capturing and Antigen Identification by LC-MS

The immuno-capturing experiment was performed with His-tagged Nb11 on *T. evansi* secretome to identify the target antigen recognized by Nb11. A total of 500 μg of *T. evansi* secretome and 25 μg of Nb11-His-tagged were mixed in a final volume of 300 μL in PBS buffer and incubated at RT for 2 h. The Nb-antigen complex was isolated using the QuickPick^TM^ IMAC kit (Bio-Nobile) following the manufacturer’s instructions. Then, 20 μL samples from each step were collected individually and analyzed on SDS-PAGE using 10% precast polyacrylamide gels (NuPAGE Bis-Tris gels, Thermo scientific) under reducing conditions and visualized by Coomassie-blue staining. The band of interest was excised from the SDS-PAGE gel and analyzed by LC-MS/MS as described [24].

### 2.6. Recombinant Production and Purification of TevENO

The gene encoding *Tev*ENO (TriTrypDB ID: TevSTIB805.10.3130) was codon optimized for expression in *E. coli*. Gene synthesis and cloning were performed by GenScript. Briefly, the synthesized gene was cloned into the pET21b vector (Novagen) using the *Nde*I and *Xho*I restriction sites. The gene was designed such to equip *Tev*ENO with an N-terminal hexahistidine (His_6_) tag and a protease cleavage site (TEV). The construct was transformed into chemocompetent *E. coli* BL21 (DE3) using the CaCl_2_ method. After transformation, single colonies were pre-cultured overnight at 37 °C with aeration (250 rpm) in 10 mL TB medium supplemented with 0.2% glucose and 100 μg mL^−1^ ampicillin. The next day, 1 mL of the pre-culture was employed to inoculate a flask containing 330 mL of TB medium supplemented with 2% glucose, 2 mM MgCl_2_, and 100 μg mL^−1^ ampicillin. This main culture was incubated at 37 °C with aeration (250 rpm) until OD_600 nm_ reached 0.6–0.8. After the induction of protein production by addition of 1 mM isopropyl β-D-1-thiogalactopyranoside (IPTG), the culture was incubated at 30 °C for 6 h with aeration (250 rpm). Cells were harvested by centrifugation (30 min, 11,000× *g*, 10 °C; Avanti J-E Centrifuge, Beckman Coulter, J10 rotor). Bacterial pellets were resuspended in 50 mL buffer A (50 mM sodium phosphate, 5 mM MgCl_2_, pH 7.0) per liter of bacterial culture. The resulting 50 mL aliquots were flash-frozen using liquid nitrogen and stored at −20 °C until further use.

Prior to purification, aliquots were thawed on ice. Cells were lysed using a sonicator (Bioblock Scientific Vibro Cell 75043; 90 sonication cycles of 5 s pulses at 20% amplitude with a 5 s pause between each cycle) and the cell lysate was centrifuged (30 min, 30,966× *g*, 10 °C). The supernatant was collected and filtered (0.45 μm). Protein purification was performed on an NGC chromatography system (Bio-Rad) using immobilized ion metal affinity chromatography (IMAC) and SEC. A 5 mL HisTrap HP nickel-sepharose column (GE Healthcare) was equilibrated with buffer A for at least five column volumes. The sample was loaded on the column using the same buffer at a flow rate of 1 mL min^−1^. After loading, the column was further washed with five column volumes of the same buffer. *Tev*ENO was then eluted by a linear gradient of buffer B (50 mM sodium phosphate, 5 mM MgCl_2_, 1 M imidazole, pH 7.0) from 0% to 100% over 20 column volumes. The fractions containing the target protein were pooled and concentrated to a final volume of 5 mL for the subsequent SEC step on a Superdex 75 16/60 column (GE Healthcare), which was pre-equilibrated with at least one column volume of buffer A. The sample was eluted at a flow rate of 1.5 mL min^−1^. Fractions containing *Tev*ENO were pooled and stored at 4 °C. Each of the purification steps was monitored by SDS-PAGE and Western blot under reducing conditions.

### 2.7. Thermal Stability of TevENO

Thermal shift assays were performed to optimize the purification and storage conditions of TevENO. The experiments were performed on a CFX Connect Real-Time System Thermal Cycler (Bio-Rad). Data were collected from 10 to 95 °C at a scan rate of 0.5 °C min^−1^. The fluorescence signal was recorded every 0.5 °C. Experiments were performed in 96-well plates and the total sample volume was 25 µL. To determine the optimal protein-dye ratio, a grid screen of various concentrations of SYPRO orange dye (Life Technologies) (0×, 5×, 10×, 50×, 100×) and TevENO (0, 1, 5, 10, 25, and 50 μM) was carried out. After identification of a suitable condition (5× SYPRO orange dye and 10 μM TevENO), buffer and additive screens were performed as previously described [43]. All experiments were performed in triplicate.

### 2.8. ELISA

#### 2.8.1. Indirect ELISA

Microtiter plates were coated with 1 μg mL^−1^
*Tev*ENO overnight at 4 °C and the excess of non-coated protein was removed by washing the plate five times with PBS-Tween20 (0.01%) (PBST). Residual protein binding sites were blocked by 300 μL blocking buffer (5% milk powder in PBS) and incubated for 2 h at RT. After washing the plate with PBST for three times, 100 μL of each Nb was loaded onto blocked wells at a concentration of 1 μg mL^−1^ and incubated for 1 h at RT. After five washing steps with PBST, 100 μL of mouse anti-HA antibody (Eurogentec) diluted in 1:2000 in blocking buffer was added to each well and the plate was incubated for 1 h at RT. The plate was subsequently washed five times with PBST, incubated for 1 h at RT with 100 μL of goat anti-mouse HRP antibody (Sigma) 1:4000 diluted blocking buffer, followed by five final washing steps with PBST. The ELISA plates were developed by addition of 100 μL TMB substrate, incubated for 10 min at RT, and stopped by adding 50 μL 1 M H_2_SO_4_. The plates were read at OD_450 nm_ with a VersaMax ELISA Microplate Reader (Molecular Devices).

#### 2.8.2. Sandwich ELISA

In the set-up of both homologous and heterologous sandwich ELISA assays, the capturing Nb was His-tagged and the detection Nb was HA-tagged. For the homologous ELISA assay, the capturing Nb was coated at varying concentration from 0 to 10 μg mL^−1^. After blocking with 5% skimmed-milk protein diluted in PBS, 100 µL of 5 µg mL^−1^
*Tev*ENO in PBS was added to the blocked wells. The detection Nb was added at varying concentration from 0 to 10 μg mL^−1^. For the heterologous ELISA assay, the capturing Nb was coated at concentration of 1 μg mL^−1^, *Tev*ENO was diluted in PBS to a final concentration of 1 μg mL^−1^, the detection Nb was added at concentration of 1 μg mL^−1^. The following steps were the same as described in Section 2.8.1.

### 2.9. SDS-PAGE and Western Blot

SDS-PAGE analysis was performed on 12% Bis-Tris protein gels (Thermo Scientific). All samples were mixed with 10× reducing agent (NuPAGE, Life technologies) and 4 × loading buffer (NuPAGE, Life technologies) and boiled at 100 °C for 5 min. The electrophoresis was performed at 120 V for 90 min. The gel was visualized by staining with Coomassie blue for 30 min and destained overnight with destaining solution (40% methanol, 10% acetic acid, and 50% distilled water). The PageRuler Prestained Protein Ladder (ThermoFisher Scientific) was employed as a molecular mass marker.

The protein gel was transferred onto a Hybond^TM^-C nitrocellulose membrane (Amersham Bioscience) for 60 min at 110 V. The membrane was blocked with 5% skimmed-milk protein diluted in PBS (blocking buffer) overnight at 4 °C. The membrane was washed with PBST for three times and incubated for 1 h at RT with mouse anti-His antibody (Pierce) 1:2000 diluted in blocking buffer. After five washing steps with PBST, the membrane was finally incubated with goat anti-mouse-HRP antibody (Sigma) diluted at 1:4000 in blocking buffer for 1h at RT. After a final washing step (five times with PBST), the membrane was developed with DAB Substrate (ThermoFisher) at RT until the bands of interest became visible on membrane, after which the reaction was halted by washing the membrane with water.

### 2.10. Circular Dichroism (CD) Spectroscopy

CD spectra were recorded on a J-175 spectropolarimeter (Jasco). Continuous scans were taken using a 1 mm cuvette, a scan rate of 50 nm min^−1^, a band width of 1.0 nm, and a resolution of 0.5 nm. Five accumulations were taken at 20 °C in buffer A at a protein concentration of 0.2 mg mL^−1^. The raw CD data (ellipticity θ in mdeg) were normalized for protein concentration and number of residues, yielding the mean residue ellipticity ([θ] in deg cm^2^ dmol^−1^) according to the following equation:(1)$$\left[\theta \right]=\frac{\theta .MM}{n.C.l}$$
where molecular mass (Da), the number of amino acids, protein concentration in mg mL^−1^, and pathlength of the cuvette (cm) are represented by *MM*, *n*, *C*, and *l*, respectively.

### 2.11. Enzyme Activity Assay

The *Tev*ENO activity was measured as previously described [44] with some modifications. Briefly, the enolase activity was measured by coupling its reaction to PYK and LDH and by following the decrease of NADH absorbance at 340 nm using a microplate reader (SpectraMax Plus 384, Molecular devices). This enzyme kinetic assay was performed at 25 °C in a 100 μL reaction mixture containing 0.1 M triethanolamine/HCl, pH 7.6, 1.1 mM ADP (Sigma A2754), 0.42 mM NADH (Sigma N8129), 2 mM MgSO_4_, and 17 mM KCl. The auxiliary enzymes PYK and LDH mixture solution (Sigma P0294) were used at 4 and 6 U mL^−1^. The substrate 2-PGA (Sigma 79480) was added as indicated.

The Michaelis constant (*K_m_*) of *Tev*ENO for 2-PGA was determined using the above-mentioned reaction conditions, by varying the concentration of 2-PGA between 0 and 1 mM. Kinetic parameters were calculated from Michaelis–Menten plots by curve-fitting of experimentally determined data using SoftMax Pro7 Software (Molecular Devices).
(2)$$V_{0}=\frac{V_{max}.\left[S\right]}{\left(K_{m}+\left[S\right]\right)}$$
where *V*_0_ = initial reaction rate (μmol min^−1^ mg^−1^), *V_max_* = enzyme maximum rate (μmol min^−1^ mg^−1^), [*S*] = substrate concentration (μM), *K_m_* =Michaelis constant (μM) of the enzyme (TevENO). The raw data (milliOD min^−1^) was transformed into specific enzymatic activity units (μmol min^−1^ mg^−1^) using the follow equation:(3)$$Specific\;activity\;(\mu \mathrm{mol}\;\min^{-1}\;{\mathrm{mg}}^{-1})=\frac{Absorbance\;decrease\;velocity\;(milli\;OD\;\min^{-1})}{ext.\;(NADH\;6220\;M^{-1}\;{\mathrm{cm}}^{-1}).\;path\;length\;(0.25\mathrm{cm}).\;[enzyme]\;(\mathrm{mg}{\mathrm{mL}}^{-1})}$$

### 2.12. Analytical SEC

The stoichiometry of *Tev*ENO-Nb11 complex was determined by analytical SEC using an ENrich^TM^ SEC 650 10 × 300 Column (Bio-Rad), pre-equilibrated in buffer A. Samples of 500 μL containing 500 μg *Tev*ENO, 168.9 μg Nb11, or *Tev*ENO-Nb11 complex mixed at varying molar ratios (2:1, 2:2, 2:3, and 2:4), were injected onto the column and eluted at a flow rate of 1 mL min^−1^. The *Tev*ENO-Nb11 complexes were mixed and incubated for 1 h prior to their application on the column. The column was calibrated with the BioRad molecular mass standard under the same conditions. The elution peaks of all chromatograms were analyzed by SDS-PAGE as described above.

## 3. Results

### 3.1. An Unbiased Alpaca Immunization Strategy Using T. Evansi Soluble Lysates Yields a Single Nb able to Recognize a T. Evansi Secretome Component

This work aimed at developing an antigen-based Nb-immunoassay for the universal detection of both Types A and B *T. evansi* strains. Hence, an unbiased alpaca immunization strategy was designed using multiple parasite isolates derived from various geographic regions covering three continents (Figure 1). An alpaca was sequentially immunized with soluble lysate preparations from *T. evansi* CAN86K (Type A), KETRI2479 (Type B), RoTat 1.2 (Type A), Merzouga I (Type A), STIB816 (Type A), and AnTAR3 (Type A) in order to maximize the chances of identifying one (or more) target antigen(s) common to all *T. evansi* strains. The resulting anti-*T. evansi* lysate Nb library was subsequently panned against *T. evansi* STIB806 secretome, resulting in the identification of three potential binders (Figure 2a). Sequence comparison of the CDR3 regions reveals that these Nbs represent three distinct families. The binders were subsequently produced in *E. coli* and purified from a periplasmic extract via a two-step purification protocol encompassing IMAC and SEC. This protocol yields highly pure Nbs, which migrate with an expected molecular mass of ≈15 kDa (Figure 2b). Finally, the purified Nbs were tested in an indirect ELISA setup, ensuring that the binders yield detectable ELISA signals against the *T. evansi* STIB806 secretome, while not cross-reacting with mouse serum components (the *T. evansi* STIB806 parasites used for secretome preparation were isolated from infected mouse blood). As a result of these tests, Nb11 was retained for further assay development, as it fulfilled both requirements (Figure 2c,d).

### 3.2. Nb11 Targets T. Evansi Enolase

Next, a combination of immunocapturing and mass spectrometry (MS) analysis was employed to determine the identity of the target antigen of the anti-*T. evansi* secretome binder Nb11. SDS-PAGE analysis of a Nb11-mediated immunocapturing experiment on *T. evansi* STIB 806 secretome reveals the presence of a single band with an apparent molecular mass of ≈55 kDa in the captured fraction (Figure 3a), suggesting that Nb11 specifically recognizes a monomeric or a homo-oligomeric target antigen. Subsequent MS analysis of the excised gel slab containing the protein band of interest identifies the target antigen of Nb11 as the glycolytic enzyme *T. evansi* enolase (*Tev*ENO; TriTrypDB *Tev*STIB805.10.3130). Importantly, *Tev*ENO is 100% conserved among Types A and B *T. evansi* strains and shows 100% amino acid sequence identity with *T. brucei* enolase (*Tbr*ENO).

To validate the interaction between Nb11 and *Tev*ENO, the latter was produced as a recombinant protein in *E. coli* (Figure 3b) and purified using a two-step purification protocol consisting of IMAC and SEC (Figure 3c,d). Given the 100% sequence identity between *Tev*ENO and *Tbr*ENO, purification buffers were initially selected based on the published *Tbr*ENO purification protocol [44]. However, in our case, we found that *Tev*ENO readily precipitates overnight in the SEC purification buffer (50 mM Tris-HCl, 200 mM KCl, 5 mM MgCl_2_, pH 8.0). This indicates a sub-optimal *Tev*ENO stability under the used conditions and renders this buffer unsuitable for long-term protein storage. Hence, thermal shift assays (TSAs) were performed in an attempt to optimize the SEC buffer, such to improve protein stability and reduce protein precipitation. As can be seen from Figure 3e, the TSA experiments led to the identification of a buffer condition (50 mM sodium phosphate, 5 mM MgCl_2_, pH 7.0) in which *Tev*ENO displays a higher thermal stability compared to the starting buffer (an increase in the apparent melting temperature T_m_ from 46.93 to 53.96 °C). Importantly, *Tev*ENO no longer precipitates after purification in this optimized SEC buffer, thereby facilitating long-term protein storage. In addition, circular dichroism (CD) spectroscopy measurements and enzyme activity assays demonstrate that the optimized purification conditions yield properly folded and active *Tev*ENO. The CD spectrum of *Tev*ENO displays the typical features of a well-folded protein composed of both α-helices and β-sheets (Figure 3f), which is consistent with the structural features of *Tbr*ENO [45]. The enzyme kinetics experiments confirm that recombinant *Tev*ENO is enzymatically active and that the determined kinetic parameters are consistent with values previously reported for *Tbr*ENO [44] (Figure 3g). The above-mentioned protein production and purification protocol results in a yield of ≈50 mg pure, functional *Tev*ENO per liter of bacterial culture.

Finally, an indirect ELISA confirms the specific interaction between Nb11 and *Tev*ENO (Figure 3h), thereby opening up possibilities for the development of Nb11-based immunoassays targeting *Tev*ENO.

### 3.3. Nb11 Can Be Employed in a Homologous Sandwich Assay Targeting TevENO

Given the dimeric nature of *Tev*ENO [44,45], the possibility of employing Nb11 in a homologous sandwich assay to detect *Tev*ENO was explored through a combination of analytical SEC and ELISA experiments.

Analytical SEC was employed to determine the stoichiometry of the *Tev*ENO-Nb11 complex through a titration. Here, Nb11 and *Tev*ENO were mixed in various molar ratios in terms of their monomer equivalents (*Tev*ENO:Nb11 monomer ratios of 2:1, 2:2, 2:3, and 2:4) and the elution profiles of the resulting complexes were compared to those collected for *Tev*ENO and Nb11 alone (Figure 4a–g). In accordance with reports in the literature [44,45], *Tev*ENO behaves as a dimer in solution (Figure 4a,g). Likewise, Nb11 elutes at a position expected for a monomeric single-domain antibody fragment (Figure 4b,g). Mixing *Tev*ENO and Nb11 in 2:1 and 2:2 ratios (i.e., one and two copies of Nb11 per *Tev*ENO dimer, respectively) leads to gradual shifts of the *Tev*ENO peak to the left, which indicates the formation of gradually larger complexes. In combination with SDS-PAGE analyses of the collected peak fractions, which clearly show that Nb11 and *Tev*ENO co-elute, these observations provide strong support for the formation of distinct *Tev*ENO-Nb11 complexes at these molar ratios (Figure 4c,d,g). Interestingly, at 2:3 and 2:4 ratios (i.e., three and four copies of Nb11 per *Tev*ENO dimer, respectively), the position of the *Tev*ENO-Nb11 complex elution peak no longer shifts indicating that the size of the complex no longer increases (Figure 4c,d,g). In addition, at these molar ratios, a Nb11 excess peak appears which becomes larger with the addition of more Nb11. Altogether, the analytical SEC data demonstrate that the stoichiometry of the *Tev*ENO-Nb11 complex is 2:2. In other words, the *Tev*ENO dimer harbors two Nb11 binding sites, suggesting that this particular system would be suitable for the development of a diagnostic test with a homologous set-up.

The potential use of a Nb11-based homologous immunoassay for the detection of *Tev*ENO was assessed through an ELISA experiment in which His- and HA-tagged versions of Nb11 (Nb11H and Nb11HA) were employed as capturing and detecting agents, respectively. The ELISA was performed in the format of a checkerboard titration in order to determine those concentrations of the Nb11 capturing and detecting variants that yield the highest ELISA signal. The highest signal is observed when Nb11H and Nb11HA are used at concentrations of 2.5 and 10 μg mL^−1^, respectively (Figure 4h).

### 3.4. Alpaca Immunization with Recombinant TevENO Yields Four Additional Nbs That Can Be Paired with Nb11 in Heterologous Sandwich Assays Targeting TevENO

While homologous immunoassays have their merits, heterologous sandwich assays are more appealing as they facilitate lateral flow assay (LFA) design, which is a potent format for the development of a rapid diagnostic test (RDT). To enable the development of a heterologous immunoassay, a new anti-*Tev*ENO Nb library was generated by immunizing an alpaca with recombinant *Tev*ENO in order to obtain more *Tev*ENO-specific binders that can be used in combination with Nb11. A panning strategy in which the library was screened against both recombinant *Tev*ENO and *T. evansi* secretome in parallel resulted in the identification of four potential additional binders (Figure 5a). Based on a sequence comparison of their CDR3 regions, these four Nbs belong to distinct families. These Nbs were produced and purified (Figure 5b) and their binding specificity towards *Tev*ENO was confirmed in an indirect ELISA (Figure 5c). A heterologous sandwich ELISA employing His-tagged Nb11 and HA-tagged variants of the newly identified *Tev*ENO binders as capturing and detecting agents, respectively, shows that three out of four (NbR1-10, NbR2-103, NbsR3-74) can be combined with Nb11 for successful *Tev*ENO detection in this heterologous format (Figure 5d).

## 4. Discussion

The ability to differentiate between ongoing and past infections represents an important difference between antigen-based and antibody-based immunoassays. The latter are often burdened by the presence of long-lasting antibodies that remain in circulation after parasite clearance, which makes it cumbersome to distinguish current from past infections [46,47]. While this issue is non-existent in antigen-based assays, this immunoassay type has its own requirements, especially with regards to the target antigen or biomarker. The target antigen should be (i) stable in the host environment and produced over the entire course of the infection, (ii) available at concentrations that allow for its detection, (iii) specific for a given pathogen, and (iv) the capture and detection antibodies targeting the antigen should be able to out-compete infection-induced host anti-pathogen antibodies [25,48,49,50,51,52,53]. Previously developed antigen-based immunoassays for *T. evansi* detection operated via an “unbiased” methodology; i.e., raising antibodies against whole parasites or preparations of specific parasite fractions without prior knowledge of the target antigen [19,54,55]. Typically, poly- or monoclonal antibodies isolated from animals infected with *T. evansi* were employed in an ELISA format to probe the sera of experimentally or naturally infected animals for the presence of *T. evansi* antigens. In these studies, the nature of the target antigen(s) or biomarker(s) remained unknown, thereby potentially hindering further assay optimization.

Here, we present the application of such an “unbiased” approach to the immunization of camelids with the aim of generating Nbs that can be employed for (i) the development of parasite-specific antigen-based immunoassays and (ii) the identification of the target antigen via Nb-mediated immunocapturing followed by MS analysis. Our group has previously adopted this strategy for the identification of novel biomarkers for the detection of active *T. congolense* infections [24,39]. The immunization of alpacas with soluble proteome and secretome fractions from *T. congolense* yielded highly specific Nbs targeting the glycolytic enzymes *T. congolense* aldolase (*Tco*ALD) and *T. congolense* pyruvate kinase (*Tco*PYK), respectively. Here, the sequential immunization of an alpaca with soluble lysate preparations from different *T. evansi* strains has led to the identification of a single binder (Nb11) targeting yet another glycolytic enzyme, *T. evansi* enolase (*Tev*ENO). Although these findings were initially surprising, it has become increasingly clear that trypanosomal glycolytic enzymes are enriched in the secreted fractions of many trypanosomes (including *T. brucei brucei*, *T. b. gambiense*, *T. congolense*, and *T. evansi*) for reasons that remain enigmatic [56,57,58,59]. Interestingly, the potential of glycolytic enzymes to serve as biomarkers for disease or infection is widely described in the literature. Examples include *Plasmodium vivax* aldolase and *P. falciparum* glyceraldehyde-3-phosphate dehydrogenase for malaria [60,61], tumor M2-pyruvate kinase for gastrointestinal cancer [62], human α-enolase for Behcet’s disease and cancer [63], neuron-specific enolase to differentiate Creutzfeldt-Jakob disease from other illnesses characterized by dementia [64], and enolase for both human and canine leishmaniasis [65]. Hence, it would seem that *Tev*ENO has all the potential to serve as a biomarker for the successful detection of active *T. evansi* infections.

While the use of *Tev*ENO as a candidate biomarker still requires thorough validation, the work presented here lays important foundations for the further development of both homologous and heterologous antigen-based immunoassays for the diagnosis of *T. evansi* infections. First, the identification of *Tev*ENO as a target antigen has allowed its recombinant production and purification, which, based on our experience with *Tco*ALD and *Tco*PYK, will be important for assay optimization and development [24,39]. Given that *Tev*ENO and *Tbr*ENO are 100% identical at the amino acid level, the production and purification conditions described for *Tbr*ENO were initially applied. As reported by Hannaert and colleagues [48], the His-tagged variant of the enzyme possesses the tendency to aggregate overnight, which in turn precludes long-term protein storage and thus significantly hampers assay optimization using recombinantly obtained *Tev*ENO. This issue was resolved in this work by performing TSAs, which led to an adaptation of the composition of the *Tev*ENO purification buffers described by Hannaert and co-workers [48]. Second, the characterization of the interaction between *Tev*ENO and Nb11 provides insights with regards to the potential flaws and strengths of the possible assay set-ups. Analytical SEC experiments reveal a 2:2 stoichiometry for the Nb11-*Tev*ENO complex (i.e., the *Tev*ENO dimer possesses two Nb11 binding sites), thereby advocating the development of a Nb11-based homologous assay targeting *Tev*ENO. A homologous ELISA performed in a checkerboard titration format to determine the optimal concentrations of capturing and detecting Nb11 shows that the highest ELISA signal is obtained with concentrations of 2.5 and 10 µg mL^−1^, respectively. Interestingly, both lowering and increasing the amount of capturing Nb11 appears to have a negative impact on the ELISA signal. This is reminiscent of the observations made for the Nb474-based homologous sandwich ELISA targeting *Tco*ALD, which could be explained by “self-competition” or “washing out” effects [66]. Whether or not similar effects are at play for the Nb11-*Tev*ENO system remains to be investigated. Third, the generation of additional anti-*Tev*ENO Nbs through alpaca immunization with recombinantly obtained, functional *Tev*ENO enable the development of heterologous assays. Despite the utility of a homologous immunoassay, heterologous sandwich assays are more easily translated from an ELISA to a LFA format [39]. In turn, this facilitates the development of an RDT that can be employed in resource-poor settings. In this respect, the NbR2-103/Nb11 pair seems especially interesting as it generates the highest signal in a heterologous sandwich ELISA targeting recombinant *Tev*ENO. Further experimentation will have to assess the performance of this Nb pair on naïve serum spiked with *Tev*ENO, serum from experimentally infected animals (and thus containing native *Tev*ENO), and field samples.

Finally, knowing the identity of the candidate biomarker generates insights on the potential breadth of the above-mentioned Nb-based homologous and heterologous immunoassays. Given that glycolytic enzymes are highly conserved among trypanosomatids, the possibility exists that the Nb-based immunoassays targeting *Tev*ENO could be employed for the detection of trypanosomes other than *T. evansi*. Indeed, this candidate biomarker would also enable diagnosis of infections with *T. brucei brucei*, *T. b. gambiense*, *T. b. rhodesiense*, and *T. equiperdum* as *Tev*ENO, *Tbr*ENO, and *Teq*ENO are 100% identical at the amino acid level. Taken that all these pathogens have a specific geographic distribution and host range, this means that a single screening tool could be employed in a one-health approach in multiple locations. For the detection of both Human African Trypanosomosis and Atypical Human Trypanosomosis outside the tsetse belt, the rapid diagnosis of trypanosomosis using an ENO-based RTD could serve as a first line screening tool. In this case, parasitological confirmation will always be a second step, before and individual treatment regimen is decided upon, after disease stage determination. In contrast, when an ENO-based RTD would be used for animal trypanosomosis, both the detection of *T. brucei* and *T. evansi* would result in the same treatment decision (individual or herd-based), depending on the host animal or geographic location, not the trypanosome species. Finally, an ENO-based positive test-score in horses would be the result of either a *T. evansi* or *T. equiperdum* infection, which would be most likely result in a PCR-based follow-up of the individual animal and a final veterinarian decision that would be instructed by the confirmation of a positive trypanosomosis score, rather than the species determination of the actual infective agent.

## 5. Conclusions

In conclusion, the work presented in this paper identifies the glycolytic enzyme Enolase as a candidate biomarker for the detection of trypanosome infections, and outines the possibility of using a Nb-based immunoassay detecting TevENO as a diagnostic for active cases of trypanosomosis. The data presented in this work provide a solid base for future validation of the test under field conditions.

## Figures and Tables

**Figure 1 vaccines-08-00415-f001:**
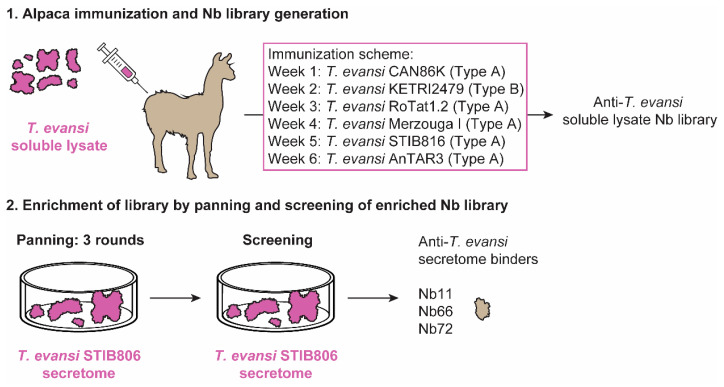
Schematic overview of the construction of the anti-*Trypanosoma evansi* lysate Nanobody (Nb) library. The anti-*T. evansi* lysate Nb library was generated from an alpaca that was sequentially immunized with *T. evansi* soluble lysates from six different strains (CAN86K, KETRI2479, RoTat 1.2, Merzouga I, STIB816, and AnTAR3) (**1**). Phage panning was performed on a secretome preparation from *T. evansi* STIB806 for three consecutive rounds. Single colonies from each round were screened for potential binders recognizing *T. evansi* secretome components (**2**).

**Figure 2 vaccines-08-00415-f002:**
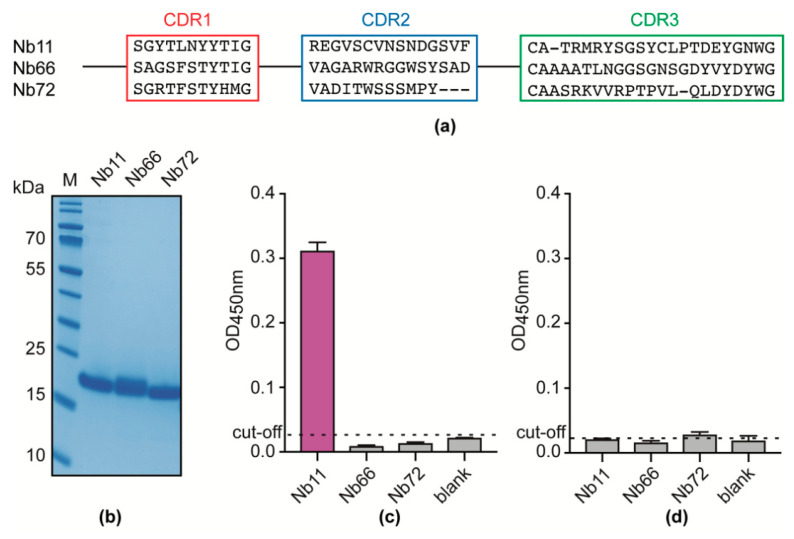
Identification of anti-*T. evansi* secretome Nbs. (**a**) Sequence alignment corresponding to CDR1, 2, and 3 regions of the anti-*T. evansi* secretome Nbs. (**b**) SDS-PAGE analysis of the identified Nbs after purification by IMAC and SEC. Lane M, PageRuler Prestained Protein Ladder. Indirect ELISA performed on *T. evansi* STIB806 secretome (**c**) and naïve mouse serum (**d**) with the retrieved Nbs. All ELISAs were performed in triplicate (error bars are shown) and the black dashed line refers to the ELISA cut-off (without Nbs as detecting agents).

**Figure 3 vaccines-08-00415-f003:**
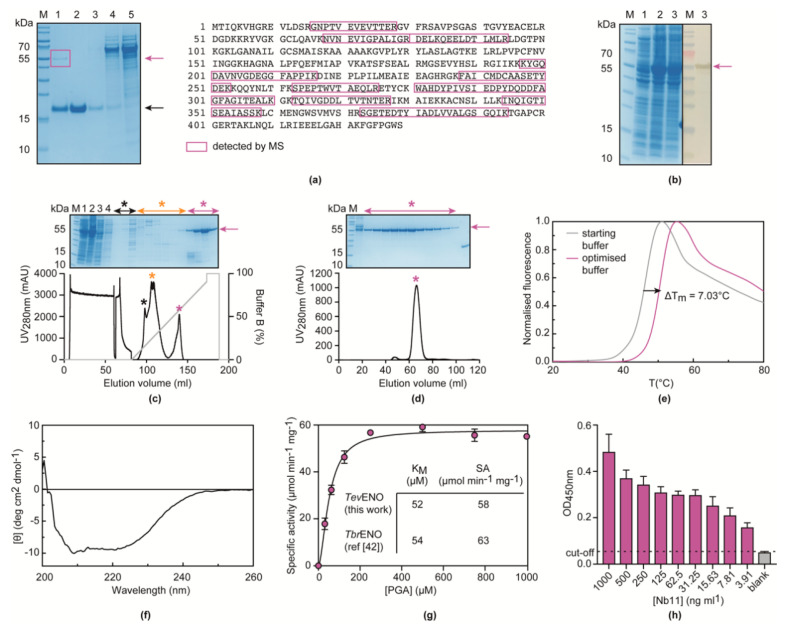
Nb11-mediated identification of *T. evansi* enolase (*Tev*ENO) as the target antigen. (**a**) Nb11-mediated immunocapturing experiment and MS analysis. *T. evansi* secretome was allowed to interact with nickel beads loaded with His-tagged Nb11. After a washing step, the captured protein was eluted, and all fractions were analyzed on a 12% SDS-PAGE developed with Coomassie blue stain. Lane M, PageRuler Prestained Protein Ladder; Lane 1, eluted fraction; Lane 2, Nb11; Lane 3, wash fraction; Lane 4, secretome flow through; Lane 5, *T. evansi* secretome (starting material). Native *Tev*ENO and Nb11 migrate at ≈55 and ≈15 kDa and are indicated by purple and black arrows, respectively. The boxed gel band was excised and sent for mass spectrometry (MS) analysis, which detected several peptides (highlighted by purple boxes) covering up to 39.86% of the entire *Tev*ENO sequence. (**b**) Recombinant production of *Tev*ENO in *Escherichia coli*. Samples of the bacterial culture were taken before induction (Lane 1), 4 h after induction (Lane 2), and 16 h after induction (Lane 3) and analyzed by SDS-PAGE (left) and Western blot (right). The band corresponding to His-tagged *Tev*ENO is indicated by the purple arrow. Lane M, PageRuler Prestained Protein Ladder. (**c**,**d**) Purification of recombinantly produced *Tev*ENO via immobilized ion metal affinity chromatography (IMAC) (**c**) and size exclusion chromatography (SEC) (**d**). The insets represent SDS-PAGE analyses of the purification procedure. Lane M, PageRuler Prestained Protein Ladder; Lane 1, supernatant of the bacterial culture after sonication; Lane 2, pellet after sonication; Lane 3, IMAC flow through; Lane 4, IMAC wash; Lanes ‘*’, various fractions collected during IMAC elution and SEC. (**e**) Results of the TSA conducted with *Tev*ENO in an effort to improve its stability during purification. The grey and purple traces represent the *Tev*ENO melting curves in the starting (50 mM Tris, 200 mM KCl, 5 mM MgCl_2_, pH 8.0) and optimized (50 mM sodium phosphate, 5 mM MgCl_2_, pH 7.0) buffers, respectively. (**f**) CD spectrum of *Tev*ENO, collected in the optimized purification buffer, shows that the recombinantly obtained enzyme is properly folded. (**g**) Enzymatic activity of recombinantly produced *Tev*ENO towards different concentration of its substrate 2-phosphoglycerate (2-PGA). The inset shows a comparison of the kinetic parameters obtained for *Tev*ENO in this work and those determined for *Tbr*ENO by Hannaert and co-workers [44]. (**h**) Titration curve of Nb11 on *Tev*ENO via indirect ELISA. The black dashed line refers to the ELISA cut-off (without *Tev*ENO).

**Figure 4 vaccines-08-00415-f004:**
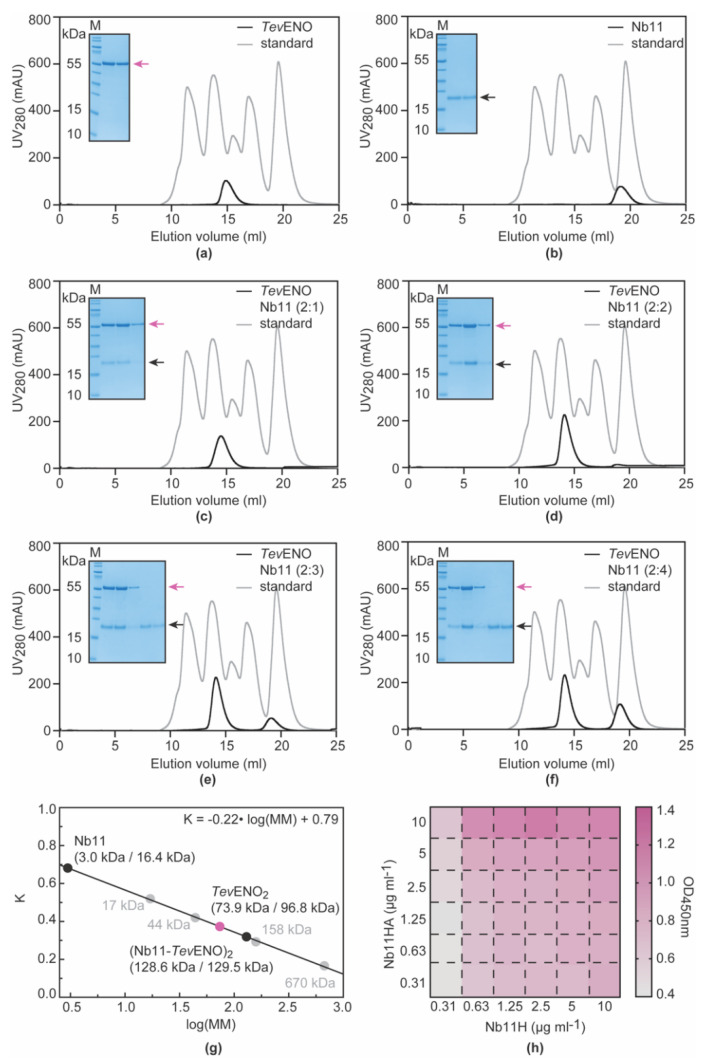
Assessment of the potential use of Nb11 in a homologous immunoassay to detect *Tev*ENO. (**a**–**g**) Determination of the stoichiometry of the *Tev*ENO-Nb11 complex by analytical SEC. Samples containing *Tev*ENO alone (**a**), Nb11 alone (**b**) and *Tev*ENO-Nb11 complexes mixed at different molar ratios: 2:1 (**c**), 2:2 (**d**), 2:3 (**e**), and 2:4 (**f**). All experiments were performed using an ENrich 650 10/30 column. The black and grey traces represent the chromatograms of the different protein samples and the BioRAD gel filtration standard, respectively. In all figures, the inset shows an SDS-PAGE analysis of elution peaks. *Tev*ENO (MM = 48.41 kDa, monomer) and Nb11 (MM = 16.35 kDa) are indicated by the purple and black arrows, respectively. *Lane M*, PageRuler Prestained Protein Ladder. (**g**) The calibration of the ENrich 650 10/30 column that allows molecular mass estimation based on the sample’s elution volume. The values between brackets indicate the estimated versus the theoretical molecular mass of the sample under investigation. (**h**) Heat map representation of the results of the Nb11-based homologous sandwich ELISA.

**Figure 5 vaccines-08-00415-f005:**
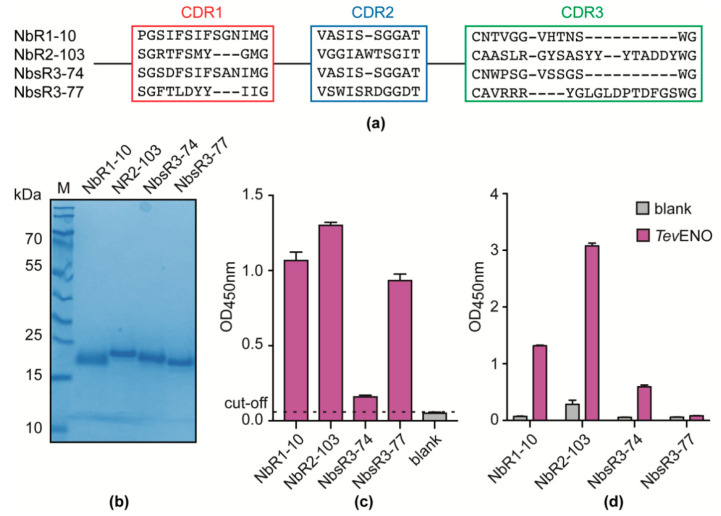
Identification of additional anti-*Tev*ENO Nbs. (**a**) Sequence alignment corresponding to CDR1, 2, and 3 regions of the anti*-TevENO* Nbs. (**b**) SDS-PAGE analysis of retrieved Nbs after purification. *Lane M*, PageRuler Prestained Protein Ladder. (**c**) Indirect ELISA performed on recombinant *Tev*ENO with the retrieved Nbs. The black dashed line refers to the ELISA cut-off. (**d**) Heterologous sandwich ELISA with His-tagged Nb11 and HA-tagged variants of the newly identified *Tev*ENO binders as capturing and detecting agents, respectively. The grey column indicates the negative control (without *Tev*ENO), whereas the purple column indicates the experimental sandwich system. All ELISAs were performed in triplicate (error bars are shown).

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
