# Peer review of "An Unbiased Immunization Strategy Results in the Identification of Enolase as a Potential Marker for Nanobody-Based Detection of Trypanosoma evansi"

_vaccines, 2020, doi:10.3390/vaccines8030415_

Round 1

Reviewer 1 Report

This study has successfully identified Trypanosoma evansi enolase as a potential maker for screening the infection through nanobody-based approaches. The study is well designed, performed well, and reported perfectly and clearly. Only a few minor revisions are necessary.

  1. L36: Species name could be abbreviated without its explanation from the second appearance after its appearance as a full spelling ‘Trypanosoma evansi’. Then, the authors could delete ‘( evansi)’ from the sentence.
  2. L143: ‘Fifty ml of …’
  3. L148-149: Delete ‘(GERBU Biotechnik GmbH, Germany)’ after ‘GERBU adjuvant’, since this explanation has been done at L140.
  4. L155, L177, L202, L236: ‘g’ in italic, when it shows standard gravity.
  5. L161: ‘One x 1011 phages particles were ….’

Author Response

  1. L36: Species name could be abbreviated without its explanation from the second appearance after its appearance as a full spelling ‘Trypanosoma evansi’. Then, the authors could delete ‘( evansi)’ from the sentence.
  2. L143: ‘Fifty ml of …’
  3. L148-149: Delete ‘(GERBU Biotechnik GmbH, Germany)’ after ‘GERBU adjuvant’, since this explanation has been done at L140.
  4. L155, L177, L202, L236: ‘g’ in italic, when it shows standard gravity.
  5. L161: ‘One x 1011 phages particles were ….’

All requested changes have been implemented accordingly.

Reviewer 2 Report

Li et al. report a method for generating Nbs that can be employed for express diagnostics and antigene identification. The manuscript is well written with solid experimental design and conclusions. It was a very enjoyable reading. I am looking forward to seeing the application of identified Nbs in diagnostics. I recommend this manuscript for publication. The only request is to improve image quality in the final version as currently some of the text in the figures is difficult to read.

i totally understand. I really think it is a solid study showing the identification and validation of the new Nbs for Trypanosoma evansi. The authors provide a detailed description of the methods sufficient for other people to reproduce, they also justify their methods. The authors also used well-established techniques, such as MS and SEC, to identify and verify antigen and antigen-antibody interaction. They showed that recombinant TevENO can yield several additional Nbs to complement diagnostic tool development. The authors even showed that new Nbs can be used for ELISA of recombinant TevENO, which is crucial for subsequent development of the diagnostic tests. Ideally, I would like to see if the new Nbs can be used for the detection of TevENO in biological fluids collected for diagnostics, moreover, the authors have access to the infected animals. Or at least discuss if the generated Nbs have sufficient binding affinities for diagnostics purposes and how sensitive the methods can be, how much simple will have to be collected, and how quick is detection. Minor comment, Figure 2 is missing description of statistics (n, error bars).

Author Response

Comment 1: “i totally understand. I really think it is a solid study showing the identification and validation of the new Nbs for Trypanosoma evansi. The authors provide a detailed description of the methods sufficient for other people to reproduce, they also justify their methods. The authors also used well-established techniques, such as MS and SEC, to identify and verify antigen and antigen-antibody interaction. They showed that recombinant TevENO can yield several additional Nbs to complement diagnostic tool development. The authors even showed that new Nbs can be used for ELISA of recombinant TevENO, which is crucial for subsequent development of the diagnostic tests. Ideally, I would like to see if the new Nbs can be used for the detection of TevENO in biological fluids collected for diagnostics, moreover, the authors have access to the infected animals. Or at least discuss if the generated Nbs have sufficient binding affinities for diagnostics purposes and how sensitive the methods can be, how much simple will have to be collected, and how quick is detection. Minor comment, Figure 2 is missing description of statistics (n, error bars).

With regards to the last part of the reviewer’s question, we have updated the legends of Figures 2 and 5 to describe the error bars and number of replicates.

With regards to testing the performance of TevENO as a biomarker when detected in biological fluids: At this point in time, we are unfortunately in a situation where we cannot perform the ideal experiment in the correct biological context. In the past, our laboratory had a permit from the university Ethics Committee to perform basic immunology studies in a T. evansi model. We no longer hold this permit and we have been unable to obtain the relevant permit in a timely manner, and have no prospect of being allowed to obtaining any new permit in the near future. In addition, the model we have used in the past is characterized by a highly virulent infection (in mice), showing parasitemia levels that can be considered not relevant for natural infection in a diagnostic setting. Moving the model into a more natural host is also impossible at our university setting. Hence, our best hope is to get the data out and we hope that through the publication we will be able to make new contacts with new research groups that have an interest in natural T. evansi infections. When this happens, we will be more than willing to share the technology with them. However, on the core issue, we wholeheartedly agree with the reviewer, which is why we intend to perform such work in the future as we have already indicated in the Discussion section (L571-573).